# SZLoc: A Multi-resolution Architecture for Automated Epileptic Seizure Localization from Scalp EEG

**Jeff Craley**[1]                                                                JCRALEY2@JHU.EDU

[1] *Department of Electrical and Computer Engineering, Johns Hopkins University Baltimore, MD*

**Emily Johnson**[2]                                                              EJOHNS92@JHMI.EDU

[2] *School of Medicine, Johns Hopkins Medical Institute, Baltimore, MD*

**Christophe Jouny**[2]                                                           CJOUNY@JHMI.EDU

**David Hsu**[3]                                                        HSU@NEUROLOGY.WISC.EDU

[3] *Department of Neurology, University of Wisconsin Madison, Madison, WI*

**Raheel Ahmed**[4]                                     RAHEEL.AHMED@NEUROSURGERY.WISC.EDU

[4] *Department of Neurosurgery, University of Wisconsin Madison, Madison, WI*

**Archana Venkataraman**[1]                            ARCHANA.VENKATARAMAN@JHU.ED

## Abstract

We propose an end-to-end deep learning framework for epileptic seizure localization from scalp electroencephalography (EEG). Our architecture, SZLoc, extracts multi-resolution information via local (single channel) and global (cross-channel) CNN encodings. These interconnected representations are fused using a transformer layer. Leveraging its multi-resolution outputs, SZLoc derives three clinically interpretable outputs: electrode-level seizure activity, seizure onset zone localization, and identification of the EEG signal intervals that contribute to the final localization. From an optimization standpoint, we formulate a novel ensemble of loss functions to train SZLoc using inexact spatial and temporal labels of seizure onset. In this manner, SZLoc *automatically learns* phenomena at finer resolutions than the training labels. We validate our SZLoc framework and training paradigm on a clinical EEG dataset of 34 focal epilepsy patients. As compared to other deep learning baseline models, SZLoc achieves robust inter-patient seizure localization performance. We also demonstrate generalization of SZLoc to a second cohort of 16 epilepsy patients with different seizure characteristics and recorded at a different site. Taken together, SZLoc extends beyond the traditional paradigm of seizure detection by providing clinically relevant seizure localization information from coarse and inexact training labels.

**Keywords:** Seizure Localization, EEG, Inexact Labels, Weak Supervision, Explainability

## 1. Introduction

Epilepsy is a heterogeneous neurological disorder characterized by recurrent and unprovoked seizures (Fisher et al., 2014) that affects roughly 1.2% of the population (Zach and et al., 2017). An estimated 20–40% of epilepsy patients are medically refractory (French, 2007) and do not respond to anti-epileptic drugs. In patients with focal epilepsy, whose seizures originate a discrete Seizure Onset Zone (SOZ) (Lüders et al., 2006), subsequent therapeutics include resective surgery and neurostimulation. Here, accurate SOZ localization is a key determining factor of treatment outcome (Rosenow and Lüders, 2001). Scalp EEG monitoring is often the first modality used in diagnosis and treatment planning for focal epilepsy. Clinical SOZ localization based on EEG relies on *visual inspection*, a process that is time consuming and often has low inter-rater agreement (van Donselaar et al., 1992).

Machine learning methods for scalp EEG have largely overlooked the challenging problem of SOZ localization, instead focusing on the simpler problem of seizure detection. Traditional approaches used a standard feature extraction and classification pipeline to categorize short windows (1–30 seconds) of the EEG as seizure or baseline (Osorio et al., 2016). More recent works have turned to deep learning via convolutional (Wei et al., 2019; Zou et al., 2018), recurrent (Vidyaratne et al., 2016; Hu et al., 2020), and hybrid CNN-RNN architectures (Craley et al., 2021; Affes et al., 2019; Liang et al., 2020). While these methods report high detection accuracies, simply determining the onset and offset times of the seizure has limited clinical utility, as it fails to provide localization information for therapeutic planning.

A few prior studies have explored the problem of SOZ localization from scalp EEG. Such methods have leveraged both cross-channel and spatio-temporal information. For example, the work of (Craley et al., 2019) uses a graphical model with related detection and localization variables to learn a spatial onset distribution for each patient. An alternative approach by (Covert et al., 2019) relies on graph convolutional networks (GCNs); the authors demonstrate that omitting EEG channels from within the SOZ greatly reduces downstream seizure prediction, suggesting a possible localization scheme based on "explainable AI". Likewise, (Dissanayake et al., 2021) posits that localization information may be revealed by analyzing the deep network trained for seizure detection. While promising, the latter two methods are geared towards a patient-specific analysis, rather than drawing inferences for new patients.

The strategy of mining cross-channel and spatio-temporal information have appeared in other EEG applications. Notably, GCN architectures have been applied to seizure detection, as in (Wagh and Varatharajah, 2020) and (Lian et al., 2020). Transformer architectures go a step further by relating sequential and spatially distributed features. The works of (Cisotto et al., 2020) and (Kostas et al., 2021) investigate transformer architectures for EEG classification tasks. In (Liu et al., 2021), spatial and temporal attention are combined into one transformer layer and evaluated on an emotion recognition task. (Qu et al., 2020) uses a convolutional feature extractor and apply a transformer for sleep staging. Finally, (Sun et al., 2021) and (Bagchi and Bathula, 2021) incorporate convolutional layers within the transformer layers for motor imagery and visual stimuli classification, respectively.

In this paper, we present the first end-to-end framework for cross-patient SOZ localization from scalp EEG. Our novel architecture, **SZLoc**, combines a convolutional, transformer, and recurrent module to efficiently learn the relevant (and heterogeneous) seizure patterns from limited training examples. Crucially, we introduce an ensemble of weakly supervised loss functions to train SZLoc to produce accurate SOZ localizations from inexact spatial and temporal onset labels. We validate the SOZ localization maps produced by SZLoc on two clinically annotated datasets. Our results demonstrate the synergy of our architecture, loss function, and training strategy for robust cross patient localization using scalp EEG. To our knowledge, this is the first result of its kind in the epilepsy literature.

## 2. Methods

Figure 1 illustrates our SZLoc architecture, which uses a multi-resolution strategy to blend information from local (electrode level) and global signal paths. Variables corresponding to each path are denoted using superscripts $e$ and $g$, respectively. Using convolutional neural networks (CNN) and transformers, global and cross-channel information informs feature

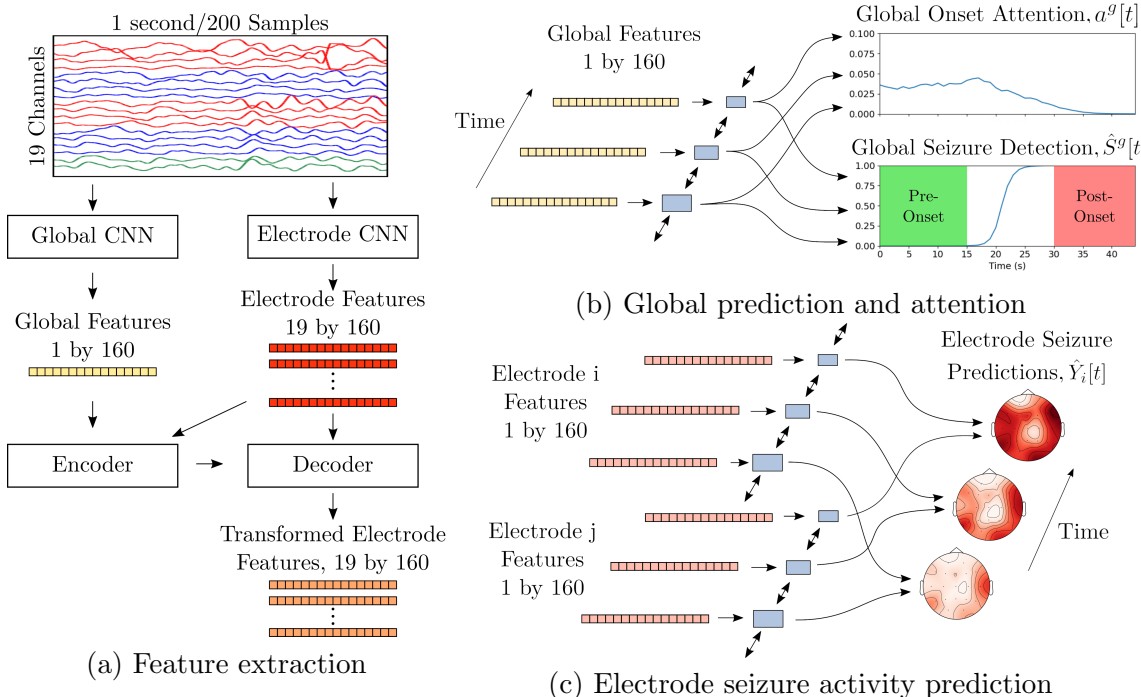

(a) Feature extraction

(b) Global prediction and attention

(c) Electrode seizure activity prediction

Figure 1: Schematic of our SZLoc framework. (a) SZLoc fuses electrode-level and global convolutional encodings using a spatial transformer to generate electrode-level representations. (b) Global features are used to generate seizure prediction $S^g[t]$ and onset attention $a^g[t]$ using GRU layers. (C) GRUs analyzing electrode features output electrode level seizure predictions $\hat{Y}_i[t]$.

extraction at each EEG electrode. The sequence of global features is used to produce seizure detection $S^g[t]$ and onset attention scores $a^g[t]$ at each time window $t$. In parallel, an electrode-wise GRU outputs the seizure activity $\hat{Y}_i[t]$ for each electrode $i$ and time window $t$. $\hat{Y}_i[t]$ is used to derive the seizure detection $S^e[t]$ and onset attention $a^e[t]$ variables. SOZ localization maps $\hat{O}^e, \hat{O}^g \in [0,1]^{19}$ are generated by combining $\hat{Y}_i[t]$, $a^e[t]$ and $a^g[t]$.

## 2.1. SZLoc Architecture

**Multi-scale Feature Extraction** As shown in Figure 1 (a), feature extraction is performed on one-second windows of the EEG. Two 1D CNNs, one acting globally across all EEG electrodes and one individually on each electrode, extract 160 dimensional features for each signal path, shown in yellow and red, respectively. A transformer layer, consisting of an encoder-decoder structure, fuses spatial information between both the global and electrode level representations of the EEG signal. Here, the encoder takes as input both the global and electrode-level features. The encoder representations are combined with the original electrode-level features in the decoder to produce the multi-scale outputs shown in orange.

**Global Seizure Activity Analysis** As shown in Figure 1 (b), global features are input into a bidirectional GRU acting temporally for seizure detection $S^g[t]$ and onset scores $a^g[t]$ for identifying the beginning of seizure. Detailed in Section 2.3, $S^g[t]$ will be trained to identify pre-seizure and post-seizure intervals. In contrast, $a^g[t]$ will be used to select time windows around the seizure onset and will be combined with the electrode-level information to generate SOZ localization maps for each seizure (see Section 2.2).

**Electrode Level Seizure Prediction** As shown Figure 1 (c), the multi-scale representations are analyzed for evolving seizure activity using a bidirectional temporal GRU. The GRU parameters are tied across electrodes to reduce model complexity. For each channel $i$ and time window $t$, the GRU outputs an electrode level detection of seizure activity $\hat{Y}_i[t]$. These detections are shown on the right of Figure 1 (c) as topographic plots on the scalp.

In training, we only have access to the onset and offset times for each seizure. Therefore, we generate a seizure detection $\hat{S}^e[t]$ for each time window $t$ by applying a max pooling operation across all electrodes: $\hat{S}^e = \max_i \hat{Y}_i[t]$. Acting as a form of weak supervision, this operation allows SZLoc to detect seizure activity at higher resolutions (i.e., channels) than the clinical annotations used for training. Onset attention scores are derived via a first order difference of the overall seizure prediction. Formally, $a^e[t] = \max\left(\hat{S}^e[t] - \hat{S}^e[t-1], 0\right)$, selecting time windows during which the predicted likelihood of seizure activity is *increasing*.

### 2.2. Generating Seizure Level Onset Maps

The datasets used in this work are recorded in the 10-20 international system, corresponding to 19 EEG electrodes. During training, we have access to clinical annotations of the *lobe* and *hemisphere* of seizure onset in each patient. This coarse SOZ information can be used to derive a binary localization map $O \in \{0,1\}^{19}$, where electrodes within the annotated lobe and hemisphere are set to 1 and the remaining electrodes are set to 0. This labeling scheme is demonstrated in Figure 2, where channels in the clinically provided annotation of "Left Temporal" are enclosed in a separate region, denoting $O_i = 1$.

The onset attentions $a^g[t]$ and $a^e[t]$ are combined with the electrode detections $Y_i[t]$ to generate a SOZ localization map $\hat{O}^g, \hat{O}^e \in [0,1]^{19}$. Here we use the generic attention $a$ and map $\hat{O}$; the superscripts $g$ and $e$ indicate global and electrode derived variables, respectively, e.g. $a^g[t]$ and $\hat{O}^g[t]$. Mathematically, the prediction for electrode $i$ $\hat{O}_i$ is computed as

$$\hat{O}_i = \frac{\sum_{t=2}^{T} a[t]\hat{Y}_i[t]}{\sum_{j=1}^{19} \sum_{t=2}^{T} a[t]\hat{Y}_j[t]} \tag{1}$$

Notice that the numerator in Eq. (1) will be large if electrode $i$ is manifesting seizure activity during the learned patient-level seizure onset time, as captured by the attention variable $a[t]$. The denominator of Eq. (1) normalizes the SOZ map, such that $\sum_i \hat{O}_i = 1$. Thus, SZLoc provides interpretable labels of seizure activity used in its final SOZ predictions.

As is standard in epilepsy monitoring, each patient may have multiple seizures. We average the predictions $\hat{O}$ across these seizures to create a patient-level SOZ map incorporating localization from each seizure. This process mirrors the SOZ information harmonization used in clinical practice, where congruent information across multiple recordings provides greater confidence in the final SOZ location. This strategy is depicted in Figure 2, where two correct and one incorrect localization combine to form a correct patient level prediction.

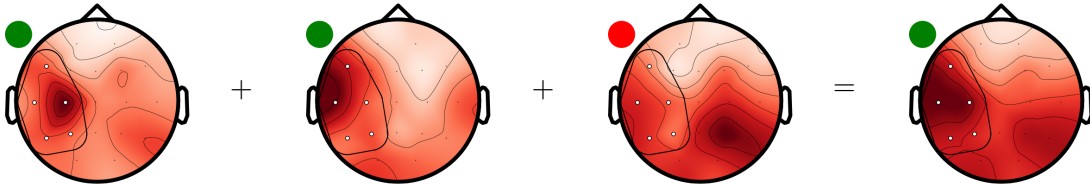

Figure 2: Localization aggregation. SOZ is correctly localized to the left temporal region in 2 of 3 seizures. After aggregation over the patient's 3 seizures, the total SOZ prediction is in the left temporal region as indicated by clinical annotations.

## 2.3. Weakly Supervised Loss Functions

Our data contains coarse and noisy detection and localization labels, from which SZLoc must learn precise, high resolution information. To learn from these labels, we separate detection and localization into separate losses and further divide these tasks based on the signal paths used. In detection, the annotated seizure interval is on a patient level and tends to be overly generous compared to the underlying electrographic signatures. To compensate, we train SZLoc on 45 s EEG snippets centered around the annotated seizure onset. Using a cross-entropy loss, we enforce that $\hat{S}[t]$ should be zero (baseline) for the first 15 s, one (seizure) for the last 15 s, and normalize over these 30 s. However, we do not enforce any label during the middle 15 s, allowing SZLoc to learn the appropriate onset time. Formally,

$$\mathcal{L}_{detection}(\hat{S}) = -\frac{1}{30}\sum_{t=1}^{15}\log(1-\hat{S}[t]) - \frac{1}{30}\sum_{t=31}^{45}\log(\hat{S}[t]) \tag{2}$$

where $\hat{S}$ may take the superscript $g$ or $e$ denoting global or electrode-level information.

On the localization front, only region-level (lobe and hemisphere) information is provided. While the SOZ lies somewhere within this region, notice that not all electrodes will necessarily manifest seizure activity at the onset. In addition seizure activity may appear outside the onset zone due to propagation effects. To accommodate these factors, we construct three complementary loss functions. Mathematically, let $\mathcal{P}(O)$ be the set of electrodes within the clinician annotated region and let $\bar{O} \triangleq \hat{O}/\max_i \hat{O}_i$ be SOZ predictions normalized such that the maximum is 1. To reward correct onset predictions in $P(O)$ and penalize those outside of it, we apply a square loss to positive and negative SOZ regions.

$$\mathcal{L}_{loc+}\left(\bar{O},O\right) = \frac{\sum_{i\in\mathcal{P}(O)}\left(1-\bar{O}_i\right)^2}{|\mathcal{P}(O)|} \qquad \mathcal{L}_{loc-}\left(\bar{O},O\right) = \frac{\sum_{i\notin\mathcal{P}(O)}\bar{O}_i^2}{19-|\mathcal{P}(O)|} \tag{3}$$

$$\mathcal{L}_{margin}\left(\bar{O},O\right) = \frac{1}{2}\left(1 - \max_{i\in\mathcal{P}(O)}\bar{O}_i + \max_{i\notin\mathcal{P}(O)}\bar{O}_i\right) \tag{4}$$

We additionally maximize the margin between regions, reflecting our evaluation strategy based on the region containing the maximum predicted electrode location.

We train SZLoc with global and aggregated electrode detection and localization losses. We apply a scaling factor to Eqs. (3-4), such that the loss terms lie in $[0,1]$, ensuring

gradients of roughly equal magnitude during training. The scaling factor for $\mathcal{L}_{loc+}$ is set to 2, while the remaining scaling factors are left at 1. The combined loss function is given by

$$\begin{aligned}
\mathcal{L}_{total} =&\mathcal{L}_{detection}(\hat{S}^g) + 2\mathcal{L}_{loc+}(\bar{O}^e, O) + \mathcal{L}_{loc-}(\bar{O}^e, O) + \mathcal{L}_{margin}(\bar{O}^e, O) \\
&\mathcal{L}_{detection}(\hat{S}^e) + 2\mathcal{L}_{loc+}(\bar{O}^g, O) + \mathcal{L}_{loc-}(\bar{O}^g, O) + \mathcal{L}_{margin}(\bar{O}^g, O)
\end{aligned} \quad (5)$$

### 2.4. Validation Strategy

We train the SZLoc architecture using leave-one-patient-out cross validation (LOPO-CV). Using this strategy, we are able to quantify generalization ability to new patients. SZLoc and competing baselines are evaluated for five random initializations and results are then averaged across these test runs. We evaluation localization accuracy based on the maximum value in the predicted onset map $\hat{O}$. If the maximum predicted weight $\arg\max_i \hat{O}$ is within the labeled SOZ for a seizure or patient $\mathcal{P}(O)$, the localization is considered correct.

**Evaluating Onset Attention** We validate our multi-signal path framework by evaluating our architectures on subsets of weakly supervised loss functions. By setting loss factors for $\hat{O}^e$ and $\hat{O}^g$ to 0, we evaluate the performance of the models using electrode and global, respectively, onset attention individually. In addition, we train the same architecture using an $\ell_2$ reconstruction loss between $O$ and $\hat{O}$, rather than the loss terms in Eqs. (3-4). Thus, we quantify the improvement due to each source of information.

**Baseline Models** Similar to the loss ablations, we evaluate SZLoc against ablated and reordered versions of the architecture. Hidden representations in each layer are of length 160, allowing modifications without the need to adjust network dimensions. The CGT baseline reorders the layers such that the GRU follows the CNN and precedes the transformer layer. SZLoc-No Connect and CGT No-Connect omit global features in the transformer input. In CG, the transformer layer is omitted, while CT omits the GRUs. SZLoc-Final State predicts SOZ using the final state of the GRU, rather than based on the attentions. Finally, the TGCN uses Architecture II of (Covert et al., 2019) modified to generate electrode level predictions. The SZLoc-Final State and TGCN models are trained using only one set of localization losses, as these models contain only one source of SOZ maps.

## 3. Results

### 3.1. Clinical EEG Datasets

Datasets of adult patients from the Johns Hopkins Hospital (JHH) and pediatric patients from the University of Wisconsin-Madison (UWM) were used in this work. Results are presented as the total number of correctly localized patients or seizures averaged over all 5 seeds. The JHH dataset contains 201 seizures from 34 adult patients with focal epilepsy and is used in our main LOPO-CV evaluation. JHH models are evaluated for generalization in the UWM dataset, consisting of 101 seizures across 16 pediatric patients. Due to differences between pediatric and adult EEG, we expect JHH model performance to degrade in the UWM dataset. Each 45 s seizure recording contains 15 s of pre-seizure, an onset period from 15 s to 30 s, and 15 s of post-onset seizure. EEG recordings are filtered from 0.5–30.0 Hz and normalized to mean 0 with STD 1, and clipped at 2 STDs. 1 s windows with no overlap were used with data augmentation as described in the appendix in Section A.5.

Table 1: JHH patient (n=34) and seizure (n=201) localizations for each training paradigm.

| Model | All Losses | | Electrode Losses | | $\ell_2$ Reconstruction | |
|---|---|---|---|---|---|---|
| | Patient | Seizure | Patient | Seizure | Patient | Seizure |
| SZLoc | $24.2 \pm 1.0$ | $109.6 \pm 8.2$ | $23.0 \pm 1.5$ | $105.2 \pm 7.0$ | $18.8 \pm 2.0$ | $104.6 \pm 12.4$ |
| CGT | $15.6 \pm 2.4$ | $83.4 \pm 10.9$ | $19.0 \pm 2.2$ | $93.6 \pm 3.3$ | $15.6 \pm 1.8$ | $88.8 \pm 2.3$ |
| SZLoc-No Connect | $19.6 \pm 2.6$ | $102.0 \pm 4.3$ | $20.8 \pm 2.9$ | $101.6 \pm 4.6$ | $17.2 \pm 2.5$ | $88.6 \pm 6.1$ |
| CGT-No Connect | $16.4 \pm 1.8$ | $81.4 \pm 7.9$ | $17.2 \pm 1.6$ | $94.8 \pm 6.6$ | $17.4 \pm 2.3$ | $94.2 \pm 7.9$ |
| CG | $19.4 \pm 2.6$ | $87.2 \pm 3.7$ | $20.0 \pm 3.0$ | $102.0 \pm 7.0$ | $16.7 \pm 1.9$ | $90.2 \pm 5.5$ |
| CT | $22.0 \pm 0.7$ | $103.0 \pm 4.8$ | $22.2 \pm 1.9$ | $108.6 \pm 4.2$ | $14.0 \pm 2.5$ | $86.4 \pm 6.1$ |
| SZLoc-Final State | $20.8 \pm 2.9^1$ | $83.0 \pm 3.1^1$ | —— | —— | $18.0 \pm 2.6$ | $91.2 \pm 10.1$ |
| TGCN | $23.2 \pm 2.4^1$ | $118.0 \pm 9.2^1$ | —— | —— | $20.4 \pm 2.0$ | $110.8 \pm 6.0$ |

Table 2: UWM generalization results for patient (n=16) and seizure (n=101) localization.

| Model | All Losses | | Electrode Losses | | $\ell_2$ Reconstruction | |
|---|---|---|---|---|---|---|
| | Patient | Seizure | Patient | Seizure | Patient | Seizure |
| SZLoc | $6.4 \pm 1.6$ | $37.8 \pm 4.4$ | $6.7 \pm 1.6$ | $39.3 \pm 4.9$ | $6.6 \pm 1.3$ | $35.4 \pm 5.1$ |
| CGT | $6.1 \pm 1.6$ | $32.1 \pm 4.5$ | $5.7 \pm 1.7$ | $34.6 \pm 5.0$ | $5.0 \pm 1.5$ | $32.8 \pm 4.5$ |
| SZLoc-No Connect | $6.5 \pm 1.8$ | $37.5 \pm 5.0$ | $6.7 \pm 1.6$ | $39.7 \pm 5.0$ | $6.7 \pm 1.4$ | $37.0 \pm 5.1$ |
| CGT-No Connect | $5.8 \pm 1.7$ | $32.7 \pm 5.1$ | $5.4 \pm 1.8$ | $33.6 \pm 4.7$ | $4.9 \pm 1.4$ | $31.9 \pm 5.6$ |
| CG | $5.9 \pm 1.7$ | $32.7 \pm 4.8$ | $5.3 \pm 1.7$ | $33.6 \pm 4.8$ | $5.2 \pm 1.5$ | $31.9 \pm 4.6$ |
| CT | $6.8 \pm 1.5$ | $38.1 \pm 4.3$ | $6.9 \pm 1.3$ | $37.2 \pm 5.0$ | $5.9 \pm 1.5$ | $32.8 \pm 5.1$ |
| SZLoc-Final State | $5.1 \pm 1.7^1$ | $31.8 \pm 4.3^1$ | —— | —— | $4.6 \pm 1.6$ | $29.2 \pm 6.3$ |
| TGCN | $5.8 \pm 1.0^1$ | $37.3 \pm 4.9^1$ | —— | —— | $5.1 \pm 1.0$ | $35.6 \pm 4.0$ |

## 3.2. Localization Results

Localization in the JHH dataset is shown in Table 1 using electrode onset attention $a^e[t]$. Performance with global attention is reported in Table 4 in the appendix. Across the board, aggregated electrode attention outperforms global attention. SZLoc outperforms all competing baselines at the patient level, achieving its best performance of $24.2 \pm 1.0$ of 34 (71.1%), though this result does not rise to the level of statistical significance due the size of the dataset. For comparison, random chance prediction would result in a patient-level accuracy of 9.6 (28.2%). TGCN achieves a higher seizure accuracy of 118.0 (58.7%) but underperforms at the patient level. Competing baselines show similar performance with no global attention applied during training, suggesting that only SZLoc is able to leverage information from both onset attention sources effectively in the multi-signal path loss paradigm. All models show performance degradation when trained with the $\ell_2$ reconstruction loss, demonstrating the efficacy of our ensemble of weakly supervised loss functions.

Table 2 shows generalization in the UWM dataset using JHH trained models. While performance is lower, all models show localization efficacy as random chance assignment would result in 4.2 (26.2%) and 25.2 (24.9%) at the patient and seizure level, respectively. Similar trends are evident, with a statistically significant SZLoc result over the TGCN. Transformer models generalize well, underscoring the benefits of the multi-scale architecture.

---

1. Baseline models with one source of attention are trained with only one set of localization loss functions.

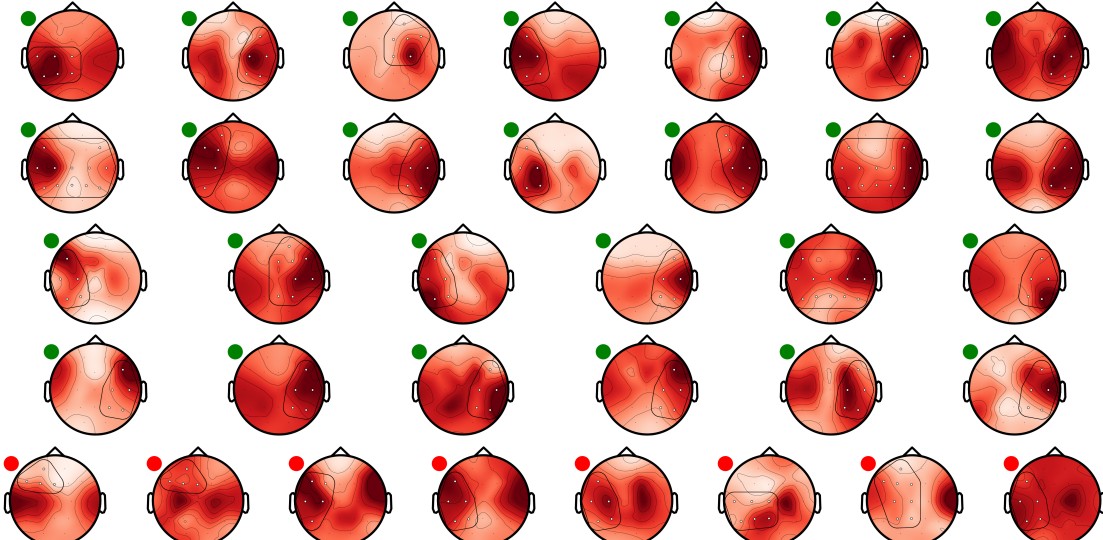

Figure 3: Patient maps $O^e$ for an SZLoc run in the JHH dataset. SOZ electrodes are shown by white circles and circumscribed in black. SOZ predictions are shown in red. Small circles indicate correct (green) and incorrect (red) localization. In 26 of 34 patients, SZLoc correctly places the highest weight within the SOZ annotation. SZLoc predicts onset near or places a second mode within $O$ in 6 missed patients.

Figure 3 shows patient SOZ localization maps for the best performing random initialization of SZLoc. SZLoc correctly identified the SOZ in 26 out of 34 patients. Secondary modes are observed when localizations for a subset of seizures diverge from the majority. This behavior conforms with clinical expectations that some seizure presentations contain better localization information than others, highlighting the benefits of patient level aggregation. We note that in most missed patients, either the mode is adjacent to the annotated region, or a secondary mode occurs within the annotated region. This demonstrates that SZLoc provides valuable clinical information even in cases where the prediction is incorrect.

## 4. Conclusion

In this paper we present SZLoc for automated epileptic seizure localization. Fusing information across multi-scale signal paths, SZLoc combines global and electrode-level spatio-temporal seizure activity leading to robust SOZ localization. An ensemble of weakly supervised loss functions allows SZLoc to compensate for coarse and noisy training labels. Namely, desirable properties of the SOZ are balanced to generate spatial maps at the seizure and patient level. By leveraging the information rich output of the SZLoc model, EEG activity contributing to the final localization can be visualized in the original signal space. We also demonstrate that SZLoc is capable of *cross-patient* localization, a relevant clinical use case. Taken together, SZLoc provides clinically useful information at multiple scales to aid in the localization of focal epileptic seizures for therapeutic planning.

## Acknowledgments

This work is supported by the National Science Foundation CRCNS award 1822575 and CAREER award 1845430, and the National Institutes of Health award R01EB029977.

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

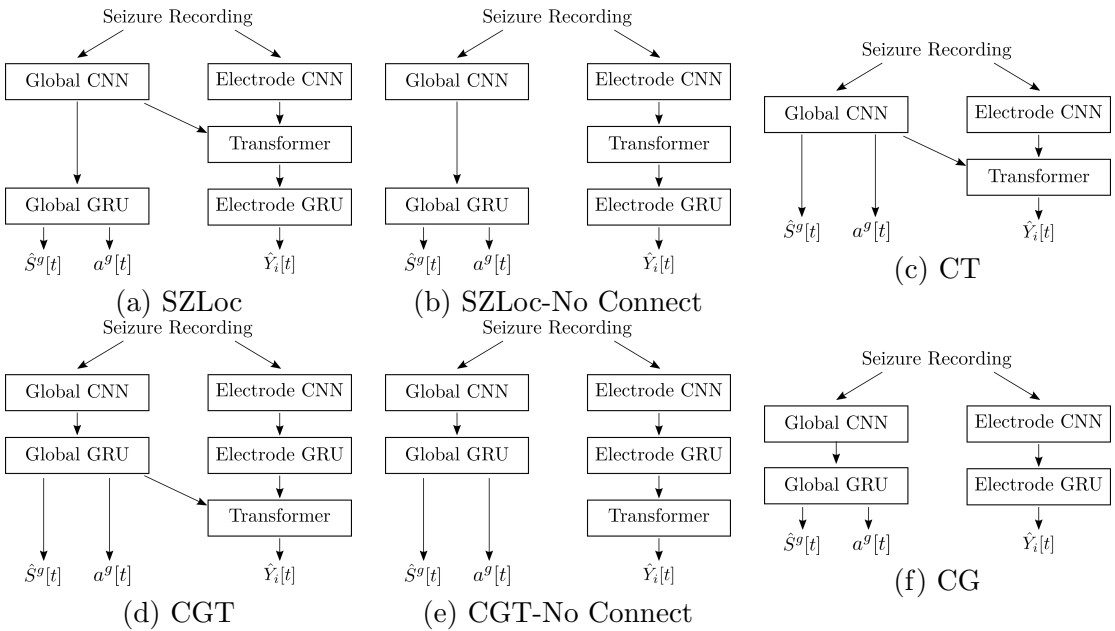

Figure 4: SZLoc (a) and ablation baseline (b–f) schematics.

Table 3: Table of symbols.

| Symbol | Meaning |
|---|---|
| $\hat{S}$ | Generic seizure detection prediction |
| $\hat{S}^g$ | Seizure detection prediction using global signal path |
| $\hat{S}^e$ | Seizure detection prediction using electrode signal path |
| $a$ | Generic attention score |
| $a^g$ | Attention score using global signal path |
| $a^e$ | Attention score using electrode signal path |
| $O$ | True onset map |
| $\hat{O}^g$ | Onset prediction score using global signal path |
| $\hat{O}^e$ | Onset prediction score using electrode signal path |
| $\bar{O}$ | Generic normalized onset prediction |
| $\bar{O}^g$ | Normalized onset prediction using global signal path |
| $\bar{O}^e$ | Normalized onset prediction using global signal path |
| $\mathcal{P}(O)$ | Set of electrodes within the SOZ region |

## Appendix A. Network Implementation Details

### A.1. Feature Extractor Implementation

A simplified schematic of the SZLoc network is shown in Figure 4 (a). We use two separate CNN architectures for the global and electrode signal paths. Each CNN consists of an embedding layer, followed by a cascade of residual blocks and convolutional projection layers. Each residual block first applies a convolution layer followed by batchnorm and a PReLU activation. Another application of convolution and batchnorm follow, with a

residual connection between the input of the block prior to the final PReLU activation. Convolutions within the block use the same number of channels as the block's input and kernels with a length 7 and a stride of 1. The projection layers between convolutional blocks double the number of channels while halving the length of the representation; they employ a kernel size of 3 and a stride of 2 unless otherwise noted. We apply global average pooling at the end of the CNN to reduce the time-series channels to a length 160 feature vector.

The global CNN starts with an embedding convolution on the 19-channel EEG signal with kernel size 7 and an output of 80 channels. Next, a residual convolution block with 80 channels is applied, followed by a projection layer to increase the number of channels to 160, followed by another convolution block. This representation is then passed through a convolution layer with 160 channels of input and output, kernel size 1, and a stride of 2 to reduce the (temporal) length of the representation. A final residual block is applied before global average pooling. We use dropout on the final feature representation during training.

The electrode CNN uses a similar architecture but increases the network depth. We train the same CNN across EEG channels to ensure consistency in the final feature representation for subsequent multi-channel analyses. The individual electrode signal is passed through an embedding convolution with kernel size 7 and 20 output channels. Next 3 residual blocks and projection layers are applied, increasing the representation from 20 to 40, 80, and 160, while halving the length of the representation after each projection. A final residual block is applied with 160 features before global average pooling and dropout are applied.

By adopting an encoder-decoder transformer structure, different sets of feature representations can be considered within each transformer component. In the encoder, global and electrode level CNN representations are used as input features while only electrode features are included in the decoder. This structure allows features from the global CNN and from each individual electrode to contribute to the final representation for each electrode. The decoder and encoder layers of the transformer follow (Vaswani et al., 2017), however only one layer is used in both the encoder and decoder. In preliminary experimentation, increased transformer depth led to poorer generalization, likely due to overfitting. The feed-forward dimension is set to 256 and dropout is applied. The 19 length 160 representations from from the electrode CNN and 160 dimensional representation from the global CNN are fed into the encoder of the transformer. The 19 electrode representations are input into the decoder, resulting in a length 160 feature representation of each electrode channel. Thus each electrode channel representation incorporates information from the other channels as well as the global EEG signal for effective multi-scale information fusion.

### A.2. Global GRU

In the GRU cell, reset and update gates, $r_t$ and $z_t$, are computed based on the previous value of the hidden state, $h_{(t-1)}$, and the input $x_t$. The sigmoid nonlinearity ensures that these gate values range between 0 and 1. The gates control how the GRU weighs (1) incoming information from the previous hidden state, and (2) the value of the new observed datapoint. An update to the hidden state $n_t$ is computed based on the input data $x_t$ and the previous state $h_{(t-1)}$, as weighted by the value of the reset gate. For example, if the reset gate is near zero, only information from the the new datapoint will be considered in the update. Finally a linear combination of the previous hidden state and the update $n_t$

are computed using the update gate $z_t$. Mathematically, the GRU is governed by

$$
\begin{aligned}
r_t &= \sigma \left( W_{ir} x_t + b_{ir} + W_{hr} h_{(t-1)} + b_{hr} \right) \\
z_t &= \sigma \left( W_{iz} x_t + b_{iz} + W_{hz} h_{(t-1)} + b_{hz} \right) \\
n_t &= \tanh \left( W_{in} x_t + b_{in} + r_t \left( W_{hn} h_{(t-1)} + b_{hn} \right) \right) \\
h_t &= (1 - z_t) * n_t + z_t * h_{(t-1)}
\end{aligned}
\tag{6}
$$

At a high level, the GRU unit weighs incoming data according to the previous values of the hidden state to produce a new hidden representation.

The bidirectional GRU in SZLoc uses 2 layers with a hidden size of 80. Thus the combined output from the forward and backward directions is 160 dimensional. To generate predictions $\hat{S}^g[t]$, the length 160 representations for each time window are fed into a linear layer with an output of 2 followed by a softmax. Global onset attention scores $a^g[t]$ are generated using a linear layer which reduces the 160 length representation to a scalar at each time window. The softmax operation is applied across time windows so that $\sum_{t=1}^{T} a^g[t] = 1$.

### A.3. Electrode GRU

The electrode GRU cells follow the same formulation as the Global GRU cells given in the previous section. Following the global signal path, the electrode signal path is fed into a 2 layer bidirectional GRU with a hidden size of 80. The 160 dimensional output is passed through a linear classification layer and a softmax is applied to generate electrode level seizure activity predictions $\hat{Y}_i[t]$. Similar to the electrode CNN, the same GRU and linear layers are trained across channels to efficiently leverage multi-channel information.

### A.4. Baseline Details

Schematics of the ablated SZLoc baselines are shown in Figure 4 (b–f). Transformer and GRU blocks are reordered and omitted depending on the architecture of the specific baseline. In addition, connections between the global features and the transformer block are omitted in the SZLoc-No Connect and CGT-No Connect baselines.

We construct the TGCN baseline according to architecture II of (Covert et al., 2019). In this network, a graph of connectivity is constructed over the EEG electrodes based on local proximity. Using this graph, a series of spatio-temporal convolutional (STC) layers are applied. In each STC layer, a graph convolution is first applied across EEG electrodes to share features between neighboring electrodes. Second, a 1D convolution is applied temporally as in a temporal CNN (Lea et al., 2016). A cascade of 8 STC layers is applied. In the original formulation, features are pooled across EEG channels for seizure detection (i.e., a single prediction for each time window). *Thus, we modify the architecture for application to seizure localization.* Instead of pooling features, we append a linear classification layer to the final nodes at every point in time to generate predictions $\hat{Y}_i[t]$ as in SZLoc.

### A.5. Data Augmentation

We use three data augmentation techniques to combat the limited size of our datasets. The first technique injects random Gaussian noise into the raw EEG signals. For each

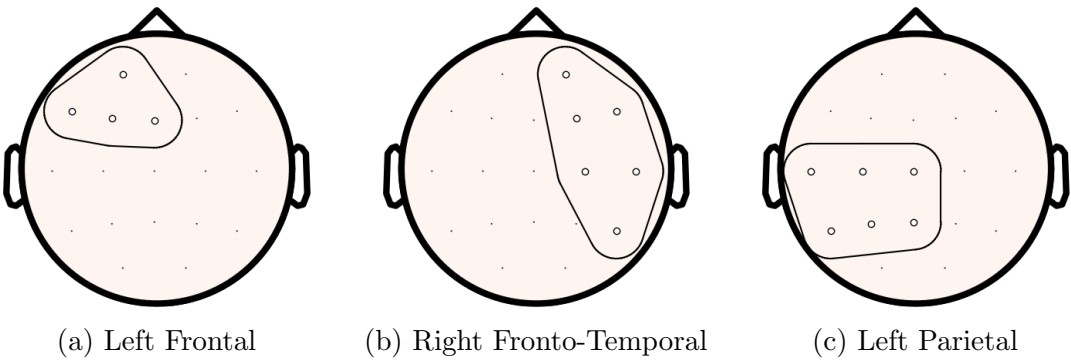

(a) Left Frontal    (b) Right Fronto-Temporal    (c) Left Parietal

Figure 5: Example seizure onset maps overlaid on the 10-20 electrodes for (a) a left frontal SOZ, (b) a right fronto-temporal SOZ, and (c) a left parietal onset zone.

EEG channel, let $M_i[t] \sim$ Bernoulli(0.5) be a random variable indicating whether or not Gaussian noise will be added. Similarly, the degree of additive noise for each window is sampled from a Gaussian distribution $V_i[t] \sim \mathcal{N}(0, 0.1)$ with variance 0.1. Mathematically, noise augmentation $G(\cdot)$ applied in channel $i$ at time $t$ can be written as

$$G(X_i[t]) = X_i[t] + M_i[t]V_i[t] \cdot \epsilon \tag{7}$$

where each entry of $\epsilon \in \mathbb{R}^{200}$ is generated from a standard normal distribution. This data augmentation technique applies noise to each channel and window individually, thus prompting robustness to changes in noise conditions across channels and time.

In addition to additive noise, signal time reversal and cross-hemispheric signal flipping is applied. For signal time reversal, the EEG signal in each window is reversed for all channels simultaneously. However, the order of the windows remains consistent, ensuring that seizure onset and propagation information is preserved. Cross-hemispheric flipping interchanges left and right channels, creating a mirror image of the original EEG signal and onset labels. This flipping ensures that the global signal path is invariant to lateralized effects. The electrode signal path remains unaffected as no sense of spatial position is preserved inherently in this part of the network. In contrast to the window based additive noise augmentation, to preserve important phase and spatial relationships between channels, both of these techniques are applied for all electrodes and time windows for a given sequence. Each augmentation technique is applied with probability 0.5 to each sequence.

## A.6. Training Details

SZLoc is implemented in Pytorch 1.9. We train the architecture for 100 epochs using Adam (Kingma and Ba, 2014), a batch size of 5, and a learning rate of 0.001. Weight decay is set to 0.001 and dropout in the CNN, transformer, and GRU are set to 0.1.

### A.7. Example SOZ Maps

Figure 5 shows example SOZ maps for different onset labels in the JHH dataset. Potential onset electrodes corresponding to the SOZ annotation are marked and encircled. In onset maps $O$, these electrodes are assigned a value of 1 while remaining electrodes are assigned a value of 0. In Figure 5 (a), electrodes corresponding to a left frontal localization are shown. Similarly, Figure 5 (b) shows an SOZ map which selects electrodes in the frontal and temporal regions as potential onset locations for a patient with a fronto-temporal SOZ. Figure 5 (c) shows onset electrodes for a patient with left parietal seizure onsets.

## Appendix B. Supplemental Results and Discussion

When comparing SZLoc to the ablated baselines in Table 1, we observe SZLoc outperforms all of them. Interestingly, the CT baseline, which contains no recurrent element, performs best among ablated baselines with a patient level electrode score of $22.0 \pm 0.7$. This result suggests that while the GRU enhances performance in the overall SZLoc architecture, when elements of the network are removed, then the GRU may overfit to the training data. This hypothesis is reinforced by the fact that the CGT baseline performs the worst of all baselines with a patient electrode performance of $15.6 \pm 2.6$. By placing the GRU before the transformer, it is likely that the model can fit to the data more directly than when the transformer occurs first in the signal path. Interestingly, when the connection between the global and the electrode signal paths are severed (SZLoc-No Connect), the performance degrades. Finally, the the CGT-No Connect performance increases when multi-scale information is not considered in the transformer, which further supports our hypothesis that placing the GRU before the transformer increases the likelihood of overfitting.

Figure 6 shows the localization maps $\hat{O}^g$ generated using the global attention variable for the same model as pictured in Figure 3. Patients are presented in the same order to allow for easy comparison between the figures. While the overall localization behavior is similar for both attention variables, we note that two patients with correct channel localizations are incorrectly localized in Figure 6. As noted in the main text, we hypothesize that the decrease in performance can be attributed to the tendency of the global attention variable to remain constant in the period before the onset of a seizure.

Figures 7 and 8 show localization maps from the two next best performing baselines, the CT and TGCN, respectively. As in Figure 6, the patients are preseted in the same order as Figure 3 for easy comparison. In its best fold performance, the CT baseline correctly identifies 23 of 34 localization zones. While the quantitative performance of the CT model is decent, the localization maps from this model are notably more diffuse, possibly due to the lack of sequence modeling. As shown in Figure 8, the TGCN correctly localizes 27 of 34 patients in its best performance, exceeding the maximum of SZLoc. However, the TGCN baseline shows a strong preference for temporal and central electrodes T7, C3, C4, and T8. The local connectivities encoded within the structured graph layers of the TGCN encourage the SOZ predictions within these electrodes, as they are common to nearly all SOZ annotations. Effectively, this allows the model to reduce the SOZ localization problem to the simpler problem of lateralization. In summary, while the TGCN baseline achieves the best quantitative performance of any of the competing baselines with $23.2 \pm 2.4$ (68.2%) at the patient level and $118.0 \pm 9.2$ (58.7%), it is apparent that the TGCN does this by overfitting

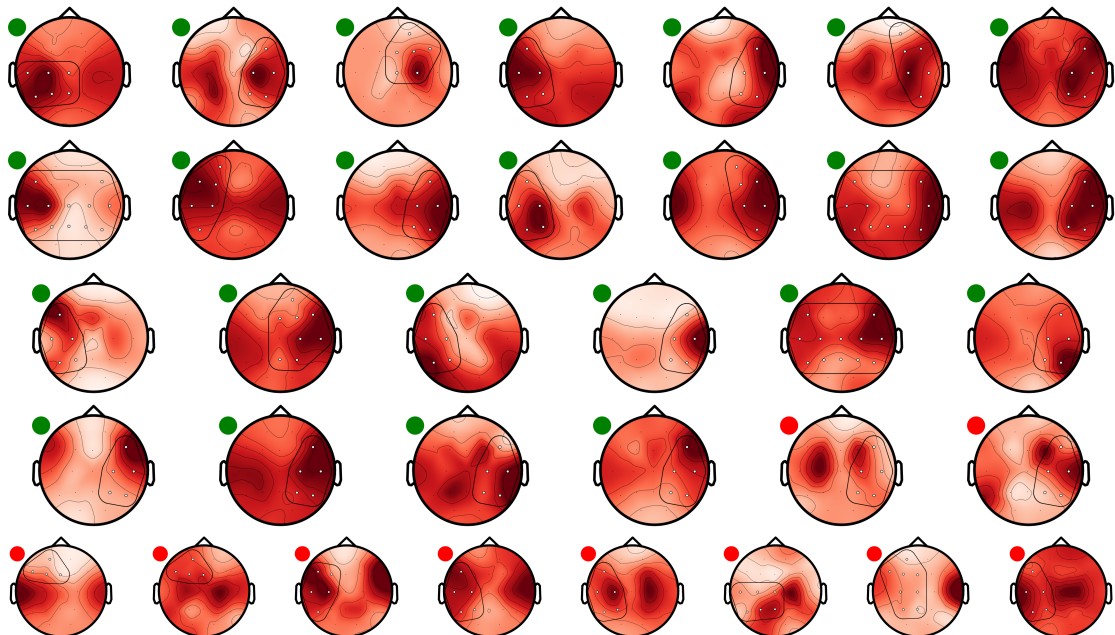

Figure 6: SZLoc global attention localization results

to the distribution of SOZs. In contrast, the spatial transformer in SZLoc does not include local connectivity information, meaning that SZLoc must learn to identify channel-level seizure activity. Hence, it is incapable of the overfitting seen in the TGCN.

Figure 9 shows seizure predictions from the (a) CT baseline and (b) TGCN baseline for the same seizure as shown in Figure 10. The CT baseline shows correct localization in the right temporal lobe. However as this model has no sequence modeling via the GRU layer, the predictions are temporally discontiguous. The TGCN, shown in Figure 9 (b), predicts almost simultaneous onset in both the left and temporal lobes. By relying on the local connectivity in the GCN layers, the TGCN baseline learns to predict seizure activity concurrently in temporal electrodes on both sides. While the TGCN achieves relatively high localization results, the model shows signs of overfitting to temporal lobe electrodes, as these electrodes are most commonly implicated as SOZ locations.

Figure 10 (a) shows predictions of seizure activity in each electrode $\hat{Y}_i$. By analyzing matrices $P$, Figures 10 (b) and (c) demonstrate how epileptic EEG signals contributing to localization can be annotated, providing added interpretability beyond SOZ map outputs. We observe that SZLoc learns to maintain a constant value for $a^g$ before fading to zero during the seizure while $a^e$ is more sharply positioned near the onset, indicating a potential explanation for electrode onset outperforming global onset in localization metrics.

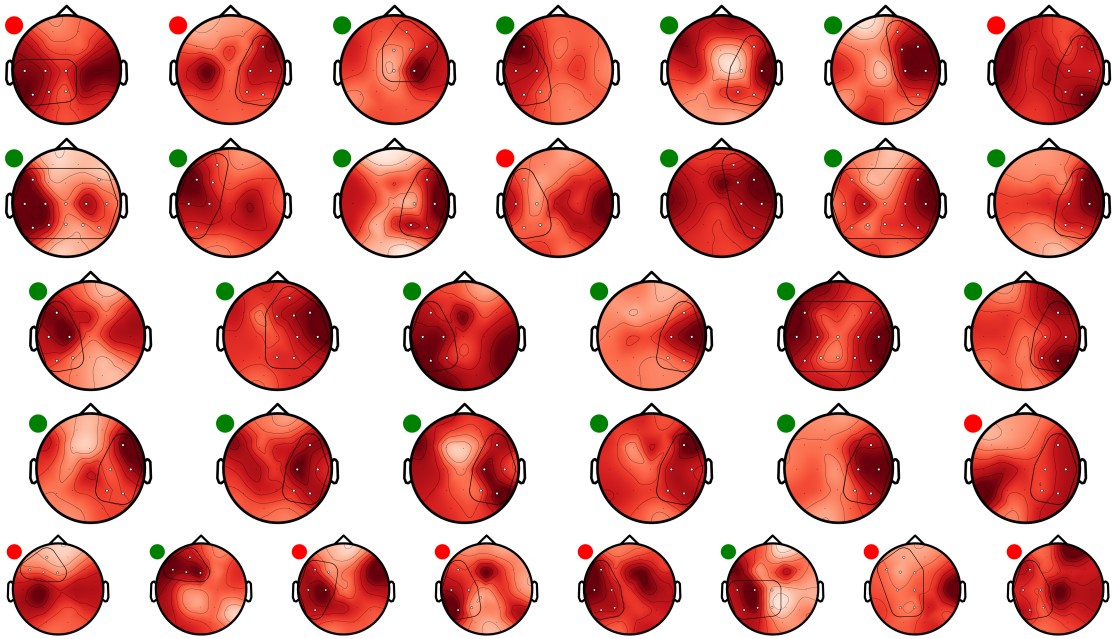

Figure 7: CT Channel attention localization results

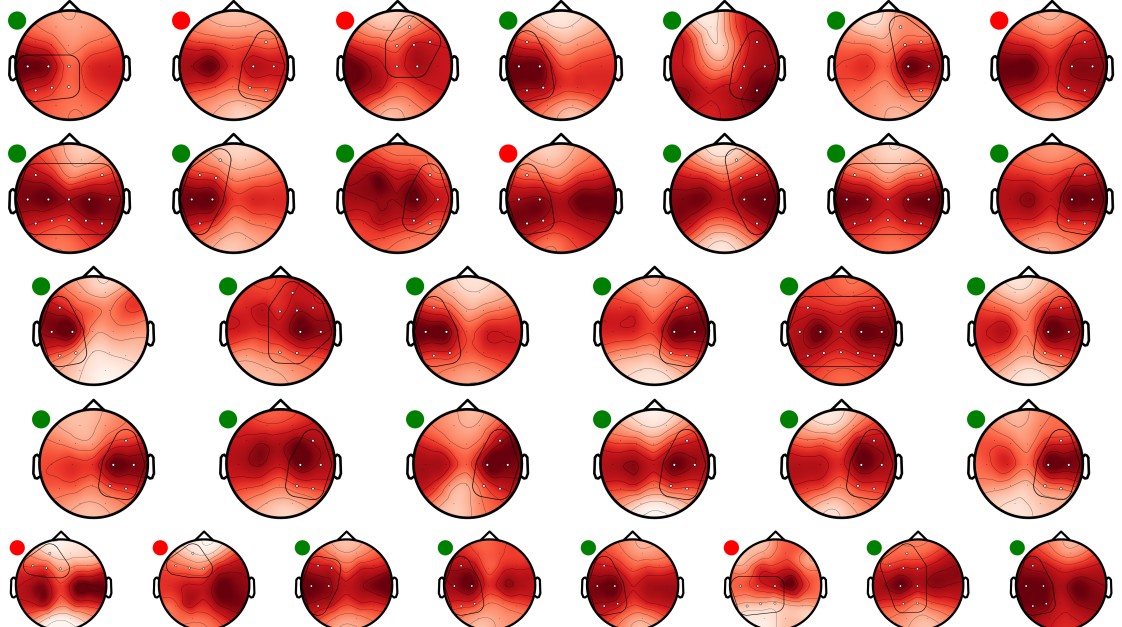

Figure 8: TGCN Channel attention localization results

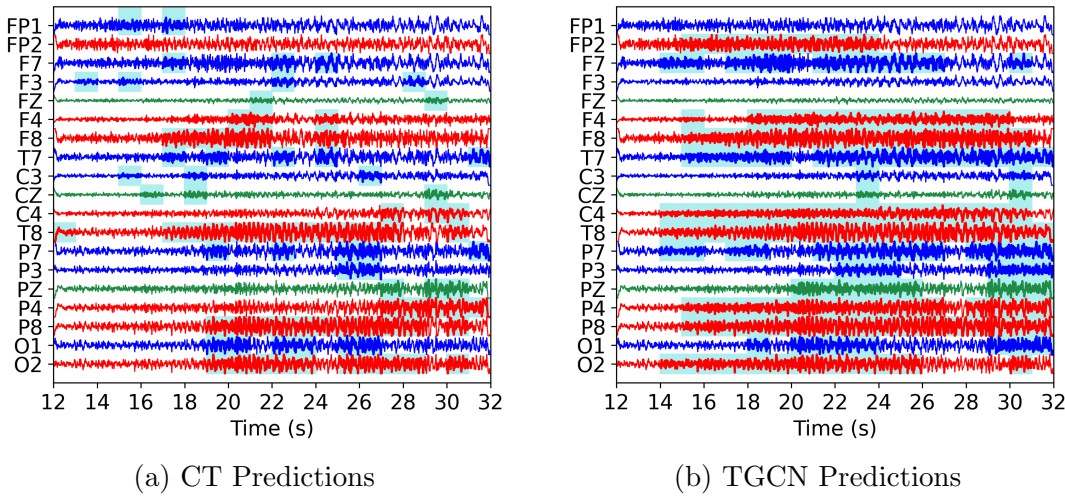

(a) CT Predictions  (b) TGCN Predictions

Figure 9: Seizure predictions from the (a) CT Baseline and (b) TGCN baseline for the seizure seizure shown in Figure 10. The CT baseline identifies the right temporal onset signal. The TGCN predicts bilateral left and right temporal onsets.

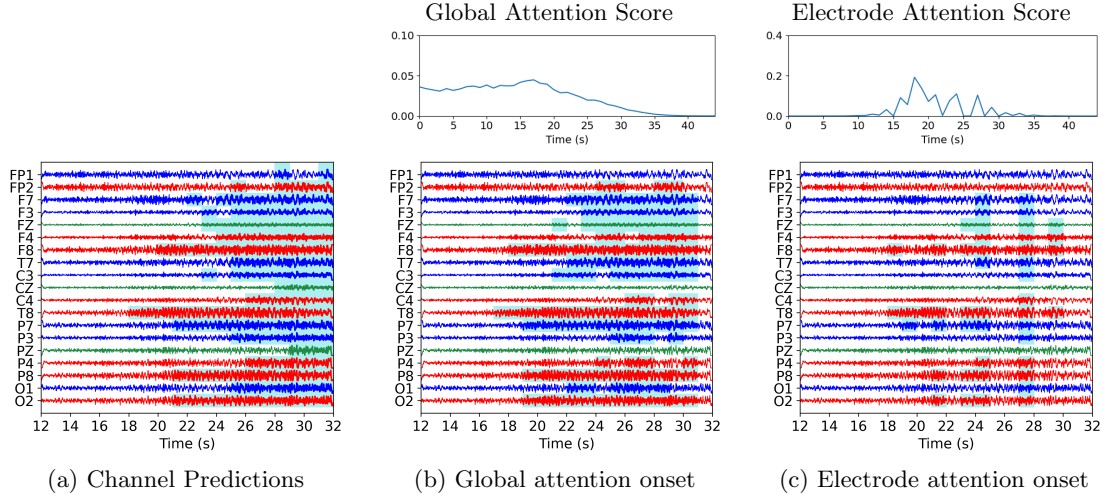

(a) Channel Predictions  (b) Global attention onset  (c) Electrode attention onset

Figure 10: Seizure and onset predictions overlayed on a EEG of a right temporal seizure. (a) Seizure predictions $\hat{Y}$ from each individual channel are shown. Seizure activity begins in channel T8 and spreads to neighboring channels. Derived variable $a^g$, $P^g$ and $a^e$, $P^e$ are shown in (b) and (c), respectively. By visualizing $Y_i[t]$ and $P$, seizure activity contributing to SOZ maps can be easily identified.

## Appendix C. Localization Performance Using Global Attention

Table 4: JHH patient (n=34) and seizure (n=201) level localization results for each training paradigm using global attention.

| | All Losses | | Global Losses | | $\ell_2$ Reconstruction | |
|---|---|---|---|---|---|---|
| Model | Patient | Seizure | Patient | Seizure | Patient | Seizure |
| SZLoc | $23.6 \pm 0.5$ | $112.0 \pm 5.4$ | $19.2 \pm 2.0$ | $101.6 \pm 3.3$ | $17.6 \pm 3.2$ | $100.2 \pm 8.1$ |
| CGT | $14.6 \pm 1.9$ | $84.6 \pm 8.6$ | $19.2 \pm 1.0$ | $92.4 \pm 12.8$ | $15.0 \pm 2.0$ | $86.2 \pm 3.0$ |
| SZLoc-No Connect | $18.6 \pm 0.5$ | $102.0 \pm 4.3$ | $17.6 \pm 1.5$ | $94.2 \pm 7.7$ | $14.8 \pm 2.2$ | $90.6 \pm 5.7$ |
| CGT-No Connect | $17.8 \pm 3.0$ | $80.6 \pm 8.9$ | $18.6 \pm 3.1$ | $89.4 \pm 1.8$ | $15.0 \pm 2.7$ | $89.0 \pm 6.0$ |
| CG | $20.2 \pm 2.2$ | $91.8 \pm 3.0$ | $17.2 \pm 2.5$ | $90.2 \pm 4.8$ | $15.2 \pm 1.4$ | $90.2 \pm 5.5$ |
| CT | $22.2 \pm 1.8$ | $111.8 \pm 11.6$ | $21.0 \pm 2.8$ | $109.0 \pm 3.3$ | $21.2 \pm 1.9$ | $108.4 \pm 3.5$ |

Table 5: UWM generalization localization results for global training paradigms.

| | All Losses | | Global Losses | | $\ell_2$ Reconstruction | |
|---|---|---|---|---|---|---|
| Model | Patient | Seizure | Patient | Seizure | Patient | Seizure |
| SZLoc | $6.7 \pm 1.7$ | $37.9 \pm 4.4$ | $7.1 \pm 1.6$ | $41.6 \pm 4.6$ | $6.8 \pm 1.6$ | $36.4 \pm 4.9$ |
| CGT | $6.2 \pm 1.7$ | $32.2 \pm 4.7$ | $5.8 \pm 1.8$ | $33.4 \pm 4.9$ | $5.7 \pm 1.4$ | $33.7 \pm 4.6$ |
| SZLoc-No Connect | $6.7 \pm 1.6$ | $38.0 \pm 4.8$ | $7.3 \pm 1.6$ | $41.8 \pm 4.6$ | $7.0 \pm 1.4$ | $37.5 \pm 5.2$ |
| CGT-No Connect | $6.0 \pm 1.8$ | $33.0 \pm 5.1$ | $5.9 \pm 1.5$ | $32.1 \pm 4.3$ | $5.7 \pm 1.6$ | $31.8 \pm 5.6$ |
| CG | $6.0 \pm 1.7$ | $33.2 \pm 4.8$ | $6.4 \pm 1.5$ | $36.5 \pm 4.5$ | $6.0 \pm 1.6$ | $33.6 \pm 4.8$ |
| CT | $7.0 \pm 1.5$ | $40.9 \pm 5.1$ | $6.4 \pm 1.5$ | $40.1 \pm 5.2$ | $6.9 \pm 1.5$ | $37.2 \pm 4.3$ |

Tables 4 and 5 report the localization performance when using global attention for seizure onset determination. As the SZLoc-Final State and TGCN models only have one source of SOZ map generation, we refer the reader to Table 1 in the main text for their respective performances. SZLoc outperforms competing baselines at the patient and seizure levels with correct localizations of 23.6 (69.4%) and 112.0 (55.7%), respectively. Localization performance degrades when the models are trained using the $\ell_2$ reconstruction loss. In cross dataset generalization performance, shown in Table 5, the SZLoc and SZLoc-No Connect baselines show strong generalization when only global loss is applied.

## Appendix D. Loss Ablation Study

Table 6: JHH patient (n=34) and seizure (n=201) localizations with loss functions ablated.

| Model | Electrode Attention | | Global Attention | |
|---|---|---|---|---|
| | Patient | Seizure | Patient | Seizure |
| SZLoc | $24.2 \pm 1.0$ | $109.6 \pm 8.2$ | $23.6 \pm 0.5$ | $101.6 \pm 3.3$ |
| No Detection | $21.8 \pm 0.4$ | $103.6 \pm 7.7$ | $21.8 \pm 1.6$ | $104.6 \pm 6.4$ |
| No loc+ | $16.2 \pm 2.5$ | $76.6 \pm 8.3$ | $13.6 \pm 2.7$ | $69.8 \pm 11.1$ |
| loc+ 1 | $21.8 \pm 1.0$ | $112.4 \pm 7.2$ | $21.8 \pm 2.7$ | $113.0 \pm 7.6$ |
| No loc- | $18.2 \pm 1.3$ | $99.2 \pm 4.6$ | $18.2 \pm 1.3$ | $99.0 \pm 3.1$ |
| No margin | $21.4 \pm 3.2$ | $98.8 \pm 7.2$ | $21.6 \pm 2.9$ | $99.2 \pm 8.3$ |

To assess the contribution of each loss function to the overall localization performance, we evaluate the SZLoc architecture with each loss function individually removed from the overall ensemble. In this experiment, global and electrode losses are ablated concurrently, i.e. for the No Detection model the total loss function is

$$
\begin{aligned}
\mathcal{L}_{total} = & \cancel{\mathcal{L}_{detection}(\widehat{S^g})} + 2\mathcal{L}_{loc+}(\bar{O}^e, O) + \mathcal{L}_{loc-}(\bar{O}^e, O) + \mathcal{L}_{margin}(\bar{O}^e, O) \\
& \cancel{\mathcal{L}_{detection}(\widehat{S^e})} + 2\mathcal{L}_{loc+}(\bar{O}^g, O) + \mathcal{L}_{loc-}(\bar{O}^g, O) + \mathcal{L}_{margin}(\bar{O}^g, O)
\end{aligned}
\tag{8}
$$

and similarly for the other modified loss functions. Table 6 reports results of this study for both electrode and global sources of attention. We note that SZLoc performs best with all loss functions applied using electrode attention $a^e$. Importantly, we observe that the overall performance degrades in nearly all cases when the seizure detection task is removed. This observation suggests that while we are primarily interested in seizure localization, training the network for an auxiliary detection task improves our generalization.

When evaluating SZLoc when each localization loss is removed, again we see that the full ensemble of losses performs best. The largest decrease in performance occurs when the positive loss function $\mathcal{L}_{loc+}$ is removed. As this loss function is directly responsible for enforcing localizations within the SOZ, this behaviour makes intuitive sense. Interestingly, with the $\mathcal{L}_{loc+}$ scaling factor set to 1 patient level performance decreases while seizure level localization accuracy increases. This indicates that at the higher scaling factor of 2, SZLoc places more confidence in correct localizations in individual recordings, allowing it to more correctly localize after aggregating over the set of a patient's recordings. Similarly, when $\mathcal{L}_{loc-}$, the loss component responsible for suppressing localizations outside the SOZ, is removed there is a similar but not as large degradation in performance. When $\mathcal{L}_{margin}$ is ablated the drop in performance is less dramatic, as $\mathcal{L}_{loc+}$ and $\mathcal{L}_{loc-}$ capture the positive and negative SOZ localization components. However we observe that the margin loss does contribute to increased performance in all metrics as well.

