# OpenReview forum: "SZLoc: A Multi-resolution Architecture for Automated Epileptic Seizure Localization from Scalp EEG"
_MIDL.io/2022/Conference — MIDL 2022_

### Official Review · Reviewer_BxEm · 2022-01-24

**Confidence:** 4
**Preliminary Rating:** 4
**Recommendation:** Poster

**Summary:**

This work presents a framework for  epileptic seizure localization from scalp electroencephalography (EEG). The framework combines the use of convolutional neural networks, a transformer and a residual network, which integrate global (all) and local signals (one channel) to extract multi-resolution feature information. It is tested in 2 different datasets showing good results

**Strengths:**

- Well written paper. Although, some parts of the multi-task learning setup are not fully clear, this should be fixable.
- Very nice state-of-the-art coverage
- Good experimental section, including ablation studies to assess the different elements in the loss, comparison to other baselines and two different datasets

**Weaknesses:**

Having multiple terms in a loss function does not necessarily imply that a multi-task learning framework is put in place. The presence of multiple supervisory signals coming from different tasks is not obvious as some of the outcomes are the result of combining elements from the network (but not well-defined tasks). The authors can refer to [1] for a precise definition of multi-task learning and Figures 2 and 3, in the reference, for a visual definition.

[1] Vandenhende, S., Georgoulis, S., Van Gansbeke, W., Proesmans, M., Dai, D., & Van Gool, L. (2021). Multi-task learning for dense prediction tasks: A survey. IEEE Transactions on Pattern Analysis and Machine Intelligence

**Deanonymize Review:**

no

**Detailed Comments:**

- It is difficult to understand how the different elements in the framework interact using only Fig 1 (i.e. how to go from (a) to (b) or (c)). The authors could consider including a sketch of the architecture, in appendix, to have a better view. This could sort the lack of clarity regarding the multi task learning setup.
- Please add a sentence where you clearly define your MTL setup. Exact number of tasks involved, which ones they are and what labels they use.

**Final Rating After The Rebuttal:**

4: Weak Accept

**Justification Of The Final Rating:**

I consider that the authors have well addressed the concerns raised by all the reviewers during the first reviewing phase.
This paper presents an interesting methodology that is worth presenting at MIDL.


**Paper Type:**

methodological development

**Questions To Address In The Rebuttal:**

- By using [1] as a reference, could you please state in which way the proposed framework can be considered as a multi-task learning framework? please detail which are the multiple tasks (outputs) of your model and the parameter sharing scheme used. having multiple terms in the loss does not necessarily translates to being multi-task.
- In Fig 2, the incorrect localization seems to still give a relatively high response in the final map. How is this handled? it would be good to have some more details.

[1] Vandenhende, S., Georgoulis, S., Van Gansbeke, W., Proesmans, M., Dai, D., & Van Gool, L. (2021). Multi-task learning for dense prediction tasks: A survey. IEEE Transactions on Pattern Analysis and Machine Intelligence

**Special Issue:**

no

---

### Official Review · Reviewer_bRcm · 2022-01-24

**Confidence:** 4
**Preliminary Rating:** 3
**Recommendation:** Poster

**Summary:**

This paper presents a deep architecture for the analysis of EEG signal of epilepsy patients. The proposed design aims at combining an efficient detection of the epilepsy seizure onset time as well at its localization. The model is based on a feature encoding part based on two CNN, one extracting a global feature vector form the whole input EEG signal (19 electrodes) and one extracting one feature vector per electrode. A transformer block is then used to put attention on the most important part of both the global and electrode signals. These two groups of features (global versus electrode-based) are then input to two GRU  : one acting on the global feature vector allows detecting the seizure onset time, a second GRU acting on the electrode-based features produces localization maps of the onset zone. This method is applied on two private (as far as I understand) datasets.



**Strengths:**

-The topic of analyzing EEG signal is interesting and quite atypical for the MIDL conference audience
-Novelty of the end-to-end architecture combining attention modules and multi-task learning exploiting both global and local information to retrieve both temporal information of the seizure onset time and its spatial localisation
-Interesting loss functions accounting for uncertainty of both the reference onset time and spatial localisation
-State-of-the art is well written and provides recent references




**Weaknesses:**

-The main weakness is that the paper is not easy to review for a non-specialist of the EEG signal processing and analysis for the following reason:
-State of the art mentions recent works in the domain of EEG signal analysis and the result section contains comparison with one model but these results are not discussed, making it difficult to evaluate the soundness of the contribution
-The main novel contributions of the authors are not clearly described: It is hard to retrieve from the synthetic state of the art to say if the proposed end-to-end architecture is original or if it is adapted from a previously proposed one. Same question regarding the proposed loss terms?

**Deanonymize Review:**

no

**Detailed Comments:**

-Some description details should be provided for non-expert in the field, eg regarding GRU, or TGCM. – Some illustrations of the different baseline models should be provided, it is indeed very hard (did not manage) to understand how the CG or CT architectures are built. Figures would really help.


**Final Rating After The Rebuttal:**

4: Weak Accept

**Justification Of The Final Rating:**

I thank the authors for their thorough review and added methodological details in the revised version of the paper. My only concern is the over length of the appendix section. I agree to upgrade my rating to weak accept.

**Paper Type:**

both

**Questions To Address In The Rebuttal:**

The authors should address comments reported in the two previous sections, especially regarding comparison with state of the art architectures for this specific application and underline the main contribution of this paper. They also should provide more details on the different compared architectures of the experimental section.

**Special Issue:**

no

---

### Official Review · Reviewer_Wcqu · 2022-01-26

**Confidence:** 4
**Preliminary Rating:** 4
**Recommendation:** Poster

**Summary:**

This paper introduces an end-to-end learning model for epileptic seizure localization from scalp EEG. The proposed method considers two representation streams, electrode-level and global-level, to compensate for different information of sources and interpretability. Regarding the weak supervision of the human annotations, they also proposed to jointly consider three complementary loss functions.

**Strengths:**

This paper considers an important and challenging problem of epileptic seizure localization from scalp EEG.
Overall, the paper is well organized, and its contributions are clear.
The experiments were conducted with two datasets to justify the validity of the proposed method.

**Weaknesses:**

The performance evaluation metric is not clearly defined.
The generalization performance on an independent dataset significantly differs from that on the training dataset.
The network architecture construction is not precise.

**Deanonymize Review:**

no

**Detailed Comments:**

It is unclear the purpose of a decoder in the transformer layer. What does a decoder DECODE? Isn’t it a kind of encoder for feature fusion?

In Eq. (2), why were the cross-entropy terms divided by 30 rather than 15?

It is hard to understand Table 1 and Table 2. What do the numbers mean? Its definition should be given.
According to the statement, “If the maximum predicted weight arg maxi Oˆ is within the labeled SOZ for a seizure or patient P(O), the localization is considered correct.”, the values are understood as accuracy (%). However, there are values larger than 100.
Why are the value ranges of “Patient” and “Seizure” different?

In terms of generalization, the values in Table 2 are much lower than those in Table 1. Hence, it is doubtful whether the proposed method is truly generalized well.

**Final Rating After The Rebuttal:**

4: Weak Accept

**Justification Of The Final Rating:**

I thank the authors' responses in detail. Accordingly, all my previous concerns have been resolved now.
Overall, this paper introduces an interesting methodology and experimental results that can draw the attention of researchers in the related field.

**Paper Type:**

methodological development

**Questions To Address In The Rebuttal:**

It is unclear the purpose of a decoder in the transformer layer. What does a decoder DECODE? Isn’t it a kind of encoder for feature fusion?

It is hard to understand Table 1 and Table 2. What do the numbers mean? Its definition should be given.
According to the statement, “If the maximum predicted weight arg maxi Oˆ is within the labeled SOZ for a seizure or patient P(O), the localization is considered correct.”, the values are understood as accuracy (%). However, there are values larger than 100.
Why are the value ranges of “Patient” and “Seizure” different?

In terms of generalization, the values in Table 2 are much lower than those in Table 1. Hence, it is doubtful whether the proposed method is truly generalized well.

**Special Issue:**

no

---

### Official Review · Reviewer_8uNc · 2022-01-31

**Confidence:** 2
**Preliminary Rating:** 4
**Recommendation:** Poster

**Summary:**

(Disclaimer: I’m not an expert in EEG. While I understand the technical aspects of the proposed architecture and will be able to comment more on that, this review will not be able to cover the ‘validation/application’ aspect of the paper.)

In this paper, a deep neural network architecture that combines CNN, GRU and Transformer layers is proposed for the task of epileptic seizure localisation - a more clinically relevant problem than simply detecting when the seizure occurred. Transformer layers are used to combine local and global features extracted by 1D CNN from the EEG electrodes and these are passed into a bi-GRU for to predict seizure activity. Global features are also passed to another bi-GRU for seizure detection (i.e. pre-seizure and post-seizure) and generation of attention scores (around seizure onset). These 3 outputs are then combined to generate localisation maps of seizure onset zones. Finally, a set of 3 weakly supervised loss functions are also proposed to tackle the problem of coarse and noisy labels provided in the dataset.

**Strengths:**

- Paper is written in a very clear structure, introduction is easy to understand even for someone without deep background in EEG.
- Validated on a separate dataset that is different from the training set (different site and seizure characteristics).
- Proposed architecture and loss functions seem well motivated + ablation studies are conducted to support the necessity of each proposed component of the architecture.

**Weaknesses:**

- Although the proposed weakly supervised loss function was compared against l2, the complexity of the loss function should warrant a more detailed ablation study, else it wouldn’t be clear exactly which of the 3 components contributed the most. Perhaps it is because of the page limit, but that could be something investigated more deeply in any follow-up work.
- Some parameters seem to be set in a rather arbitrary fashion (e.g. why is only L_{loc+} = 2? How would it perform if it was left at unity like the other scaling factors?)
- SZLoc with all losses had the best performance, but it would have been better to state whether the difference between its performance with the other models (e.g. TGCN) was statistically significant. Not having a statistically significant difference won’t simply discount the value of the work, but it will give a clearer idea of how much improvements it actually brings. (or it could be mentioned that the high SD is likely to be due to the small dataset)
- More crucially, there does not seem to be much discussion on why the performance gap between SZLoc and the models tested in the ablation study (e.g. CT, SZLoc-No Connect) become much smaller.

**Deanonymize Review:**

no

**Detailed Comments:**

- It should be mentioned (at least a short sentence in Section 3.1) in the main text that data augmentation was performed
- Eqn 2: should add a line to explain why it is being divided by 30, doesn’t seem very intuitive especially when it only sum over 15 time points
- “by setting the scaling factors for the corresponding loss terms to 0” -> would be better to mention exactly which ones were set to 0.

Minor comments
- Fig 1(a), the ‘orange’ multi-scale outputs looks too similar to the ‘red’ electrode features, might be a good idea to choose a different colour.
- Might be better to explain the difference between \hat{Y}_i[t] and Y_i[t] in Figure 1’s caption, else it’s confusing when reading the paper in the order it’s presented for the first time.
- The architecture is quite a huge one and you’re dealing with spatiotemporal data. Will be better to be very clear about which part of your architecture captures spatial information and which parts capture temporal information (even within temporal, there is a time series within the 1 second time window, and the broader pattern across multiple seconds).
- Although it might be intuitive, better to explain that superscript e refers to electrode-level while superscript g refers to global level
- The notations get rather confusing (O, \hat{O}, O^g, O^e ; \hat{a}^g[t], \hat{a}^e[t], ...), it might be useful to have it summarised in a table placed in the supplementary materials.

**Final Rating After The Rebuttal:**

4: Weak Accept

**Justification Of The Final Rating:**

In their rebuttal, the authors have reported results for an additional set of ablation study to clarify the effects of each term in the loss function. This gives a stronger defence of their proposed architecture and produced additional insights to the model performance. I'd be willing to upgrade the rating to 'Accept' if there were such an option (I'm not in the field of EEG, so I don't think I'm in a position to give it a 'Strong Accept'). Nevertheless, the current version of the manuscript provides a robust analysis of their proposed methods and it should be of interest to the other researchers in the field.

**Paper Type:**

both

**Questions To Address In The Rebuttal:**

- Is there a reason behind the rather frequent appearance of secondary modes in the SOZ maps (Figure 3)? While it is mentioned that several wrong predictions had a secondary mode within the correct area (and thus provide valuable clinical information), what would the secondary node mean in the case where the model predicts correct (i.e. the ‘primary’ mode is in the correct area, but there is a significant secondary mode outside of the correct area, e.g. 2nd row, 3rd map from the right ; 1st row, 1st map from the right)?

**Special Issue:**

yes

---

### Meta-Review · Area_Chair_2wSA · 2022-02-18

**Recommendation:** Accept (Poster)
**Confidence:** 5

**Metareview:**

The work applies a neural network (CNN + Transformer + GRU) trained with 3 losses, to process EEG signal and predict (localize) when a seizure occurred.

Overall, the reviewers identified the following primary strengths and weaknesses:

Strengths:
* Interesting problem.
* Generally well written.
* Evaluated on 2 datasets, different for train and test.

Weaknesses according to reviewers:
* Some issues with clarity regarding description of methodology and evaluation.
* Harder to understand for readers without expertise on EEG.

During the rebuttal/discussion period and the accompanied updates to the manuscript, the authors did a good job in addressing various concerns of the reviewers, clarifying various points, and these were substantiated with corresponding additions and improvements to the paper. The quality of the paper has improved after the updates. As reflected in the scores, reviewers seem to agree that the paper is of quality acceptable for publication.

---

### Decision · Program_Chairs · 2022-02-28

Accept